# Geochemistry of Brine and Paleoclimate Reconstruction during Sedimentation of Messinian Salt in the Tuz Gölü Basin (Türkiye): Insights from the Study of Fluid Inclusions

**Anatoliy R. Galamay** [1], **Muazzez Çelik Karakaya** [2], **Krzysztof Bukowski** [3,*], **Necati Karakaya** [2] and **Yaroslava Yaremchuk** [1]

1    Institute of Geology and Geochemistry of Combustible Minerals NAS of Ukraine, 79053 Lviv, Ukraine
2    Department of Geological Engineering, Engineering and Natural Sciences Faculty, Konya Technical University, 42250 Konya, Türkiye
3    Faculty of Geology, Geophysics and Environmental Protection, AGH University of Science and Technology, 30-059 Kraków, Poland
*    Correspondence: buk@agh.edu.pl

**Abstract:** The halogenesis of the Messinian Tuz Gölü Basin corresponds to the sulfate type and the magnesium sulfate subtype. Compared to the Messinian Sea brines, they have a slightly higher $[Na^+]$ concentration, which is 96.6–116.4 g/L, and a much lower $[K^+]$ concentration, ranging from 0.1 to 2.35 g/L. During salt sedimentation, the $[Mg^{2+}]$ concentration ranged from 6.1 to 14.0 g/L, and the $[SO_4^{2-}]$ concentration from 18.2 to 4.5 g/L. Physical–chemical reactions in the basin's near-surface and bottom waters during the suspension of halite deposition had a decisive influence on the significant reduction of $[SO_4^{2-}]$ sedimentation brines. During these periods, there was an intensive influx of $Ca(HCO_3)_2$ into the sedimentation basin and the formation of glauberite layers. The formation of the glauberite resulted from the slow dissolution of pre-deposited finely dispersed metastable minerals—gypsum, sodium syngenite, or mirabilite. In fluid inclusions in the halite, the sulfate minerals being allogenic crystals of calcium sulfate, are represented by gypsum, bassanite, and anhydrite. Additionally, as the other sulfate minerals, glauberite, anhydrite, and thenardite are found within halite crystals. Sharp fluctuations in daytime air temperatures characterized climatic indicators of the summer period in the Tuz Gölü region: 15.6–49.1 °C. In the spring or cool summer–autumn period, the daytime air temperature in the region ranged from 15.7–22.1 °C, and in late spring and early summer, it ranged from 20.6 °C to 35.0 °C. During some periods, the Tuz Gölü halite crystallized at 61.8–73.5 °C. The extreme high-temperature crystallization regime at the bottom of the salt-bearing basin was achieved due to the emergence of a vertical thermohaline structure. The "greenhouse effect" in the Tuz Gölü was established only briefly but was periodically renewed due to the influx of "fresh" waters.

**Keywords:** fluid inclusion; halite; Tuz Gölü basin; Central Anatolia; Miocene

## 1. Introduction

Due to the significant diversity of the conditions of ancient salt accumulation, which sometimes significantly differ from those in modern times, the question of paleogeography of ancient evaporite basins, in many cases, continues to be debatable. An important paleogeographic task is to clarify the nature of the connection of ancient salt basins with each other and with basins of normal salinity and to determine the impact of these connections on the lithology and mineral composition of salt-bearing formations.

By studying fluid inclusions in halite, which contains microdroplets of water from ancient sedimentary basins, we can solve many problems in the climatology, mineralogy, and sedimentology of some regions [1–4]. The results obtained, in comparison with the salinity data of modern lakes, provide a basis for further physicochemical modeling of the

processes occurring in ancient basin systems with semi-arid climates and give impetus to understand the causes of these characteristics.

One of the evaporite formations of Eurasia with a debatable origin is the Upper Miocene Katrandedetepe Formation, which is widely distributed in central Anatolia in the Tuz Gölü Basin (Türkiye). Previously, the isotopic composition of several minerals, such as dolomite/magnetite (C, O, Mg) [5], gypsum/anhydrite (Sr, S, O) [6,7], and halite (Cl, Br, B, Li) [8], was studied to determine the origin of the evaporites and sedimentary conditions in this basin. The chemical composition of the water extract of halite and partly the chemical composition of sedimentation solutions in halite were also studied [9]. These studies established that the brines of the southeastern part of the Tuz Gölü Basin were mainly of marine origin and that the inflow of seawater into the sedimentary basin occurred due to transgressions. It is determined that the wide range of variation in mineral isotopic composition indicates the involvement of volcanic emanations, hydrothermal solutions, and meteoritic or groundwater/river water in forming the chemical composition of the basin brines. Moreover, it is confirmed by earlier isotopic studies [5,7,8], indicating dry sedimentation conditions and significant temperature fluctuations.

In this paper, new results of chemical and microthermal analyses of primary fluid inclusions in the halite and mineralogical and petrographic studies of salt from several boreholes in the Tuz Gölü Basin were carried out. Ultramicrochemical, thermometric, immersion, crystal-optical, and X-ray spectral microanalysis methods were used. Three main objectives of this study are to establish, based on the obtained data:

(1) the direction and causes of changes in the chemical composition of brines of the Tuz Gölü Basin,
(2) the stability of the vertical zoning of the water column of the salt basin depending on climatic, geological, and hydrographic conditions,
(3) features of the paleoclimate of the region.

## 2. Geological Setting

The Tuz Gölü Basin is located in the southern part of the Konya closed basin and covers some sub-basins (Haymana–Bala and Ereğli–Ulukışla). The basin is the largest (62,000 km$^2$) inner basin among the Central Anatolian Cenozoic basins (Çankırı–Çorum, Yozgat, and Sivas). The basins completed their evolution during the Cretaceous–Eocene period and were formed due to different environments and mechanisms [10,11] (Figure 1). The Tuz Gölü basin sediments contain various compositions and characteristics, formed from the Upper Cretaceous to the Quaternary [12]. In the late Cretaceous–Eocene period in the Ereğli–Ulukışla sub-basin, which is situated in the southeastern part of the Tuz Gölü Basin, sedimentary extensional processes were active [11].

Central Anatolia started to rise due to the compression regime during the closure of the northern branch of Neotethys in the middle–late Eocene. Evaporitic rocks were deposited with gradual shallowing as a result of uplift with a structural compression regime in the Paleogene period in Central Anatolia [13]. Various types of evaporite minerals (e.g., Ca-sulfates and carbonates) and/or submarine clastics were deposited due to disconnection from the sea of the basins. These evaporites mainly formed as a result of the regressive process that widely took place in basins such as Tuzgölü, Sivas, and Ulukışla during the late Eocene–Oligocene period [14,15]. In the Ereğli–Ulukışla sub-basin, an anhydrite unit (so-called Zeyvegediği anhydrides) was formed under an external marine lagoon environment [16–18]. It is stated that anhydrite is quite common in this basin; sandstone–limestone, green marl, and brown clayey gypsum are deposited in the upper parts. The thickness of the unit is about 900 m and gradually decreases toward the west.

Akgun et al. [19] reconstructed the paleotopography of the late Eocene and late Miocene of the Ereğli–Ulukışla sub-basin based on examining palynological features of samples taken from surface outcrops. Researchers explained that as a result of the uplift of the Central Anatolian Plateau, high paleotopographic conditions were formed. As a consequence of the rise of the Taurus Mountains, the paleoclimate changed to a humid-

temperate, tropical state, and freshwater lakes were formed in the Burdigalian–Tortonian. During the Messinian Salinity Crisis (MSC) recorded in the Mediterranean basin, they explained that the basin transformed into a dry lake plain under strong, dry, and warm-temperate climatic conditions during the Messinian [19].

As the Mediterranean reached its highest level during the middle Miocene, the southern parts of the Tuz Gölü Basin were covered with seawater ([5] and references therein). Marine sediments were deposited in the Adana, Manavgat–Aksu, Mut–Ermenek, and Beyşehir regions, located approximately 50–100 km S–SW of the Ereğli–Ulukışla sub-basin ([20,21] and references herein (Figure 1)). The marine sediments in the mentioned areas indicate that the Mediterranean covered approximately 100 km from the coastline to the inner parts of Anatolia during the middle and late Miocene [22–25].

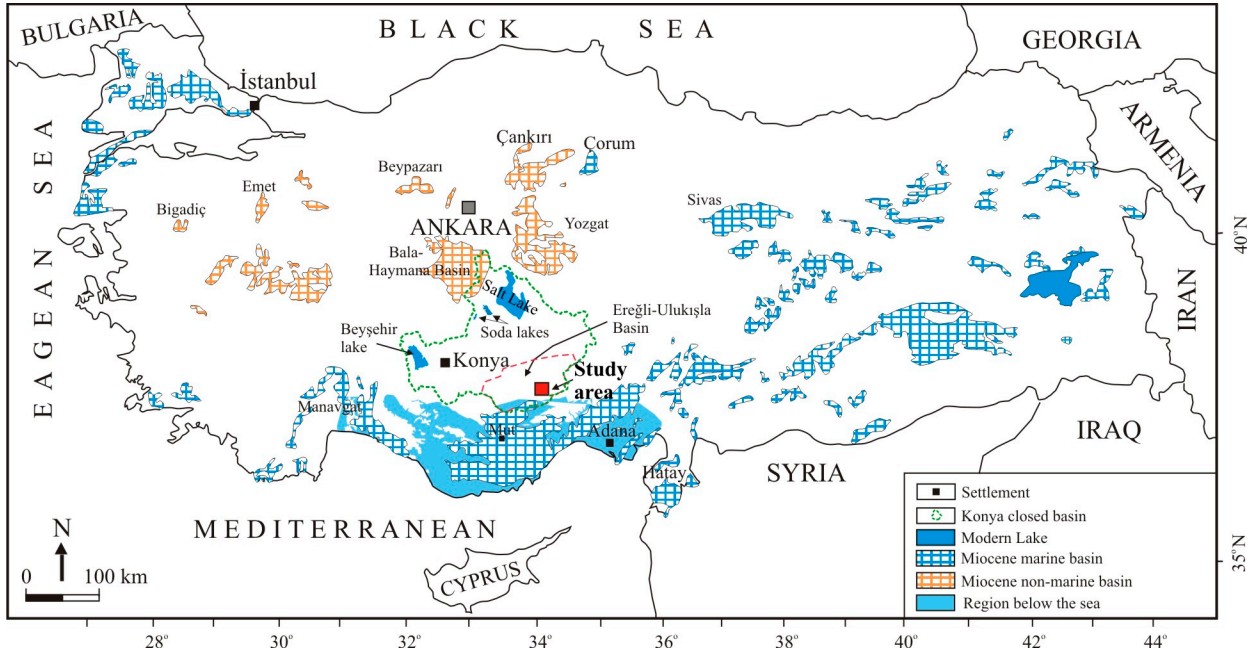

**Figure 1.** Study area on the Paleogeographic map (before 8 Ma). Pale blue areas indicate regions of modern topography that were below sea level prior to 7 to 8 Ma (modified from [26] and Miocene marine and non-marine basins in Türkiye are simplified after MTA [27].

However, by examining the drilling samples of the Ereğli–Ulukışla sub-basin, it was stated that the basin was of a lake lagoon type with partial sea water inflow during the late Miocene–Pliocene period [5,8,9]. In the Central Anatolia Basin, the sediments of the Tuz Gölü and various size soda lakes are mainly composed of halite, gypsum, Na–Ca-sulfates (mostly glauberite and thenardite), partially mirabilite, hexahydrate, and konyaite). The Pliocene–Quaternary succession consists of carbonates (calcite, dolomite, huntite), clay-rich deposits (smectite, sepiolite, palygorskite), and rarely sand–clay-sized clastic sediments [9].

## 3. Materials and Methods

Salt samples were taken from TG5, TG6, and TG7 boreholes, located 2.5–5 km from each other between Çukurkuyu, Yeniköy, and Badak villages. Their lithological sections with sampling locations are presented in Figure 2.

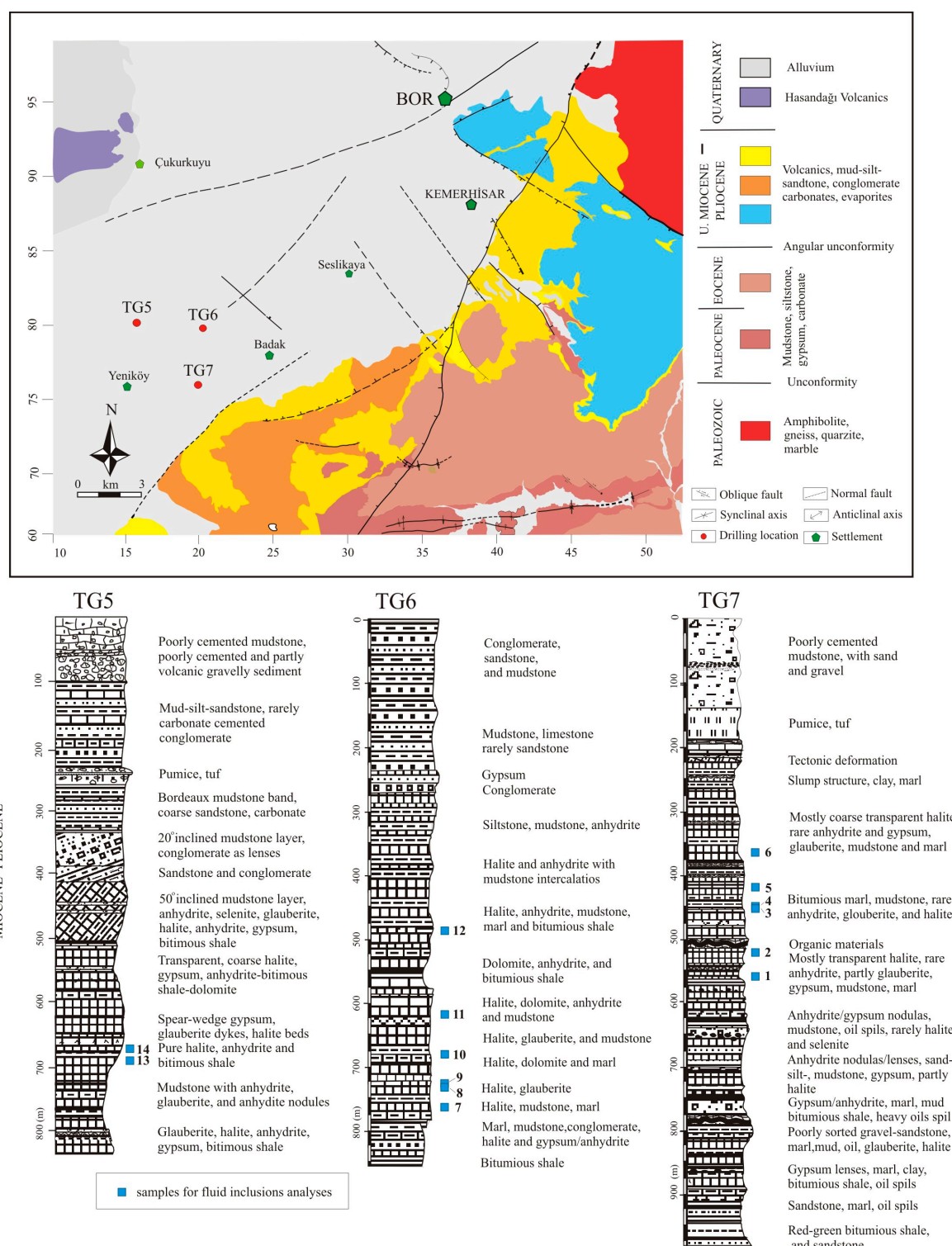

**Figure 2.** Lithological profiles and locations of TG-5, TG-6, and TG-7 boreholes on the geological map of the southern part of the Tuz Gölü Basin.

The chemical composition of brines extracted from fluid inclusions of 14 samples of halite (Figure 2) was studied via an ultramicrochemical method (UMCA). The UMCA method [3] is based on the traditional chemical analysis where inclusion brine is extracted using a glass capillary. Then, a reagent is added to determine the substances dissolved in the inclusion fluid.

The content of potassium, magnesium sulfate and calcium ions in the brines was determined. The minimum concentration of these ions for a correct determination should be, respectively: 0.8 g/L for [K$^+$]; 1.0 g/L for [Mg$^{2+}$]; 0.5 g/L for [SO$_4{}^{2-}$], and 0.9 g/L for [Ca$^{2+}$] [28]. The analytical error of a single determination is 1–24% for [K$^+$] (increases at low ion concentrations); 1–6% for [Mg$^{2+}$] in Na–K–Mg–Cl–SO$_4$ brines; 1–19% for [Mg$^{2+}$] in Na–K–Mg–Ca–Cl brines; 2–8% for [SO$_4{}^{2-}$]; 1–26% for [Ca$^{2+}$] in Na–K–Mg–Ca–Cl brines [3]. Errors decrease when 2–3 analyses are performed for each ion in a brine of the one large inclusion or different inclusions of a specific zone (band).

The data from the analysis of the chemical composition of the brines are transferred from the ionic form to the chemical formula according to the Bunsen method. This method is more convenient for many constructions and applications and allows calculating the concentration of [Na$^+$] and [Cl$^-$], e.g., [29,30].

Bromine contents of pure or halite-enriched samples were analyzed with an ELAN 6100 inductively excited plasma mass spectrometer (ICP-MS) with an Optima 7300 DV inductively excited plasma optical emission spectrometer (ICP-OES) at the AGH University of Science and Technology, Poland.

Calcium sulfate minerals trapped within fluid inclusions were determined using the immersion method, e.g., [31]. The immersion method of determining an index of refraction consists in comparing the index of a mineral with those of a series of liquids, with the final result that the index of the mineral is equal to that of one of the series of liquids or lies between those of two consecutive liquids.

All our operations were carried out under an optical microscope. The halite surface was dissolved by a thin stream of water above the selected inclusion containing the daughter crystal. The inclusion was then opened, and the small crystal was extracted with a thin glass capillary. The refractive indices of the immersed crystals and the immersion liquid preparations were compared under a MIN 8 polarizing microscope. The mineral composition of non-halite layers and the mineral composition of the insoluble halite residue were determined by an X-ray diffractometer (XRD) on an ADP-2.0 diffractometer (Fe radiation, filtered Mn, 30 kV, 12 mA). The scanning rate of the powder preparations was 2°/min, and the step was 0.025°. Quantitative mineral content was determined using Profex-8.4 software, which simulates the calculated diffractogram profile as close as possible to the experimental one [32].

The fluid inclusion homogenization temperature was obtained via a thermometric method using a thermal chamber designed by Kaluzhny [33]. That instrument allowed for the observation of the homogenization processes of a large number of inclusions in halite. With the application of such a heating chamber, research practice made it possible to establish the temperature determination accuracy of $\pm 2$ °C, in the range from 30 to 250 °C [33]. The gas phase in single-phase fluid inclusions was inducted artificially by cooling halite preparations for 2–4 days to 0–10 °C [34]. The variation of temperature values during repeated measurements did not exceed 0.1 °C; the standard heating rate was 1 °C/min.

## 4. Results

### 4.1. Sedimentary Textures of Halite

Numerous sedimentary textures of so-called "cumulate halite" and "chevron halite" were found in the halite. It was formed, respectively, in the near-surface and bottom layers of the brine. Both halite textures are well preserved (Figure 3). The size of the cumulate halite reaches 1.3 mm. Some halite crystals showed zonal texture in the bottom condition.

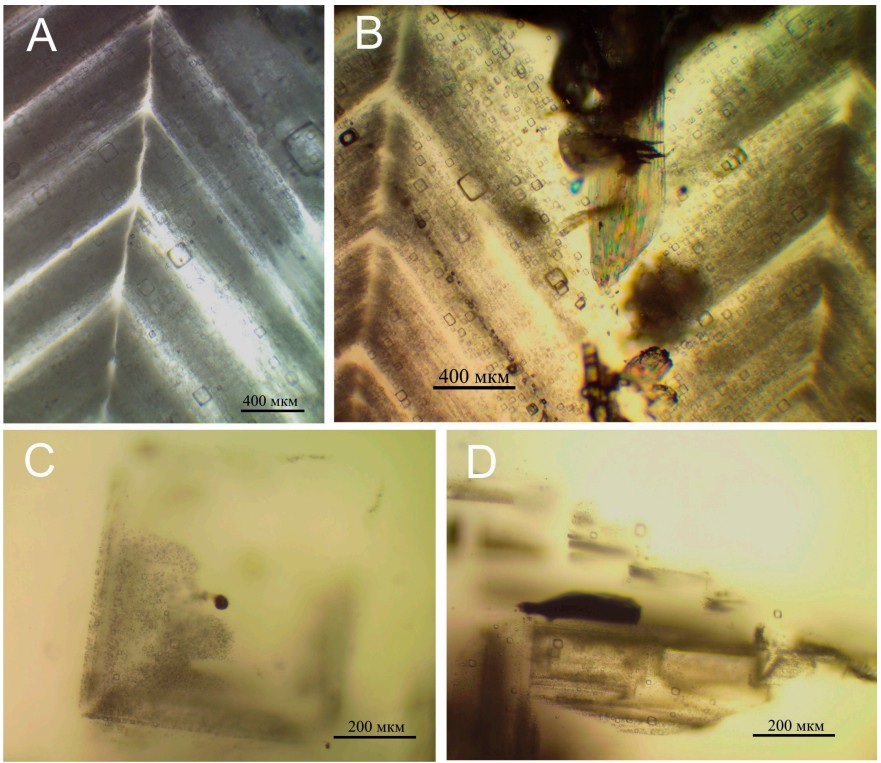

**Figure 3.** Sedimentary textures of halite from Tuz Gölü Basin. (**A**) Chevron: sample No. 7; (**B**) Chevron: sample No. 2; (**C**) Cumulate: sample No. 13; (**D**) Cumulate: sample No. 5; For sample location, see Figure 2 and Tables 1 and 2.

Bottom (chevron halite) textures are formed by separate bands of a dark color (enriched with fluid inclusions) and light color (without inclusions), which form a distinct cyclic pattern (Figure 3C), and a maximum of 13 rhythms can be distinguished. The width of the dark bands varies from 200 to 400 μm, sometimes reaching 800 μm, and the width of the light bands is usually around 50 μm. The size of the structures along the axis reaches 1.5 cm.

The textures of the so-called "salting-out halite" [3] have not been established.

Primary fluid inclusions contained in sedimentary textures are in the form of negative cubic crystals ranging in size from a fraction of one μm to 500 μm. Inclusions are single-phase (liquid), two-phase, and multiphase. Multiphase inclusions can contain a gas phase, terrigenous clay material, allogenic crystals of various minerals, and organic material in the form of oil or bituminous matter, algae, spores, and pollen (Figure 4). The internal pressure in the inclusions is close to normal atmospheric pressure.

The cumulate halite captured the myospores on the brine's surface and then transported them to the bottom of the basin. The accumulation of spores and pollen of some plant species in various inclusions indicates the staged sporulation of different plants in the region. Among the terrestrial plants that grew near the saline basin are Carya, Taxodium, Pinus sp., and others (see Figure 4).

### 4.2. Chemical Composition of Primary Fluid Inclusions in Halite

To determine the content of ions in the basin waters, we extracted brines from primary single-phase liquid and gas–liquid fluid inclusions of the bottom (chevron) halite. It was established that the content of $Ca^{2+}$ ions in primary brines during an increase in salinity was less than 0.5 g/L, which is below the sensitivity of the analysis. The $K^+$ content in the basin brines ranged from 0.9 to 2.35 g/L and, in some periods, was below 0.5 g/L, i.e., below the sensitivity of the analysis. $Mg^{2+}$ content in the brines ranged from 6.1–14.0 g/L and $SO_4^{2-}$ 4.5–18.2 g/L (Table 1).

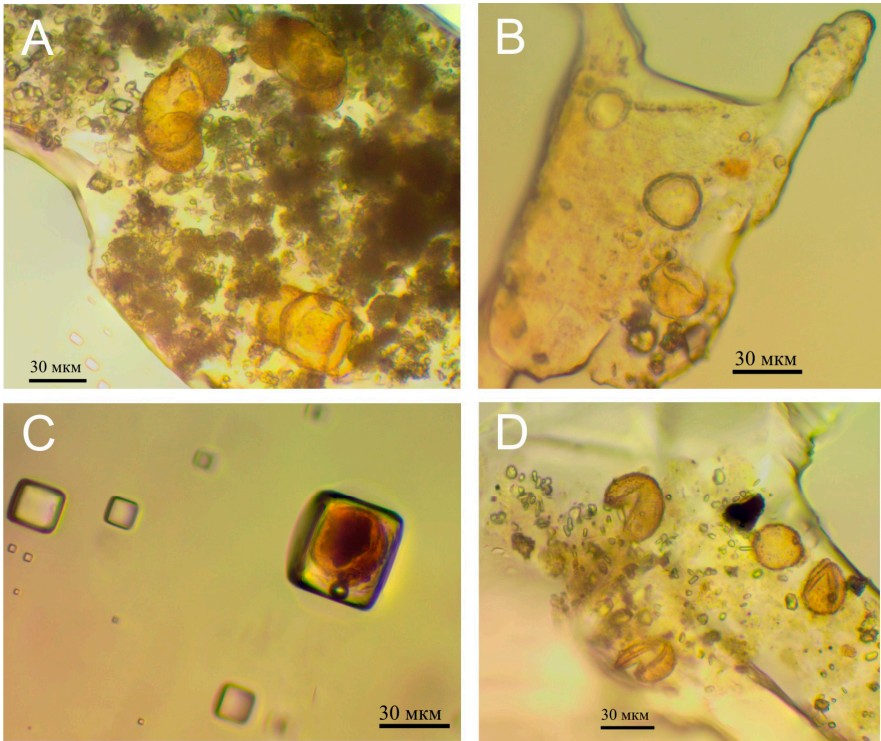

**Figure 4.** (**A–D**) Well-preserved micro-phytoplankton, numerous spores, and pollen of terrestrial plants in primary and secondary fluid inclusions in the halite of sample No. 11 (well TG 6, depth 618.4 m).

According to studies of the crystallization of easily soluble salts during the gradual concentration of natural brines, their saturation with NaCl and halite precipitation are achieved at the appropriate total ion concentration and solution density [35]. The dependence of $Cl^-$ and $Na^+$ content on the mineralization of natural brines is manifested as follows: the content of $Cl^-$ at the stage of precipitation of halite (420 g/L before mineralization) remains stable and amounts to 180–192 g/L, while the range of $Na^+$ changes with a trend of progressive decrease according to the deposition of halite [35,36].

To calculate the actual content of sodium and chlorine in the brines of the Tuz Gölü Basin, we relied on the data obtained from a study of the physical and chemical properties of natural brines. The error in the determination of chlorine and sodium is 7%. Therefore, the sodium content in the sedimentation solution of the Tuz Gölü ranged from 96.6–116.4 g/L, and the ratio of $r$Na/$r$CI was 0.80–0.97 (Table 1).

**Table 1.** The chemical composition of Tuz Gölü brines, and Cenozoic marine brines presented as salts (in the brackets sum of ions).

| Sample | Location, Well-Depth of Sampling, m | Content, g/L | | | | $r$Na/$r$CI |
|---|---|---|---|---|---|---|
| | | **NaCl** | **KCl** | **MgSO₄** | **MgCl₂** | |
| 14 | TG5-677.9 | 270.4 (106.4 + 164.0) | 0.2 (0.1 + 0.1) | 9.9 (2.0 + 7.9) | 29.4 (7.5 + 21.9) | 0.88 |
| 13 | TG5-889.8 | Fluid inclusions are smaller than 40 μm | | | | |
| 12 | TG6-484.9 | 266.9 (105.0 + 161.9) | 2.7 (1.4 + 1.3) | 13.8 (2.8 + 11.0) | 30.6 (7.8 + 22.8) | 0.87 |
| 11 | TG6-618.4 | 289.9 (114.1 + 175.8) | 2.5 (1.3 + 1.2) | 22.8 (4.6 + 18.2) | 12.1 (3.1 + 9.0) | 0.95 |
| 10 | TG6-679.5 | Sample without primary fluid inclusions | | | | |
| 9 | TG6-728.5 | 276.8 (108.9 + 167.9) | 1.9 (1.0 + 0.9) | 5.6 (1.1 + 4.5) | 23.1 (5.9 + 17.2) | 0.90 |
| 8 | TG6-730.0 | 295.8 (116.4 + 179.4) | 2.4 (1.25 + 1.1) | 22.4 (4.5 + 17.9) | 7.4 (1.9 + 5.5) | 0.97 |
| 7 | TG6-762.7 | 258.7 (101.8 + 156.9) | 1.7 (0.9 + 0.8) | 10.6 (2.1 + 8.45) | 38.0 (9.7 + 28.3) | 0.85 |

**Table 1.** *Cont.*

| Sample | Location, Well-Depth of Sampling, m | Content, g/L | | | | rNa/rCI |
|---|---|---|---|---|---|---|
| | | NaCl | KCl | MgSO₄ | MgCl₂ | |
| 6 | TG7-360.3 | 283.4 (111.5 + 171.9) | 3.4 (1.75 + 1.6) | 9.1 (1.8 + 7.3) | 16.8 (4.3 + 12.5) | 0.92 |
| 5 | TG7-422.0 | 270.6 (106.5 + 164.1) | 2.7 (1.4 + 1.3) | 13.0 (2.6 + 10.35) | 27.7 (7.05 + 20.6) | 0.89 |
| 4 | TG7-450.0 | 279.1 (109.8 + 169.25) | 1.8 (0.95 + 0.85) | 8.4 (1.7 + 6.7) | 21.4 (5.45 + 15.9) | 0.91 |
| 3 | TG7-455.0 | 275.3 (108.3 + 167.0) | 1.7 (0.9 + 0.8) | 20.3 (4.1 + 16.2) | 24.5 (6.25 + 18.25) | 0.90 |
| 2 | TG7-518.6 | 280.1 (110.2 + 169.9) | 0.3 (0.15 + 0.1) | 5.9 (1.2 + 4.7) | 21.5 (5.5 + 16.0) | 0.91 |
| 1 | TG7-562.9 | 245.5 (96.6 + 148.9) | 4.5 (2.35 + 2.1) | 10.0 (2.0 + 8.0) | 47.0 (12.0 + 35.0) | 0.80 |
| | Eocene seawater at the stage of halite crystallization, after [37] | | | | | |
| IX | Navarra, Spain | 122.8 (48.3 + 74.5) | 31.3 (16.4 + 14.9) | 15.7 (3.2 + 12.5) | 129.7 (33.1 + 96.6) | 0.40 |
| | Badenian seawater at the stage of halite crystallization, after [38] | | | | | |
| VIII | Carpathian area | 211.9 (83.4 + 128.5) | 15.4 (8.1 + 7.3) | 22.1 (4.5 + 17.6) | 67.4 (17.2 + 50.2) | 0.73 |
| | Messinian seawater at the stage of halite crystallization, after [37,39–41] | | | | | |
| VII | Red Sea Basin | 167.3 (63.8 + 98.3) | 29.2 (15.3 + 13.9) | 57.5 (11.6 + 45.9) | 99.1 (25.3 + 73.8) | 0.60 |
| VI | | 212.6 (81.6 + 125.7) | 15.3 (8.0 + 7.3) | 31.3 (6.3 + 25.0) | 71.3 (18.2 + 53.1) | 0.71 |
| V | Caltanissetta, Sicily | 101.0 (48.3 + 74.4) | 34.1 (17.9 + 16.2) | 65.2 (13.2 + 52.0) | 171.6 (43.8 + 127.8) | 0.49 |
| IV | Lorca, Spain | 186.4 (71.0 + 109.4) | 25.2 (13.2 + 12.0) | 50.9 (10.3 + 40.6) | 86,7 (22.1 + 64.6) | 0.65 |
| | Modern seawater at the stage of halite crystallization, after [35,36] | | | | | |
| III | Bahamian Islands (W46) | 255.9 (103.0 + 152.9) | 7.4 (3.9 + 3.5) | 22.1 (4.5 + 17.6) | 31.7 (8.1 + 23.6) | 0.90 |
| II | Bahamian Islands (W33) | 215.2 (84.2 + 131.0) | 16.6 (8.7 + 7.9) | 47.6 (9.7 + 38.2) | 71.3 (18.2 + 53.1) | 0.72 |
| I | Black Sea | 262.5 (104.1 + 158.4) | 6.3 (3.3 + 3.0) | 26.3 (5.3 + 21.0) | 40.0 (10.2 + 29.8) | 0.90 |

In order to compare the Tuz Gölü brines and Cenozoic marine brines (in terms of age and chemical composition of different concentrations), the percentages of individual ions were calculated (according to the conversion of g/L content into mol%), and the results were given in Table 2.

**Table 2.** Chemical composition of Tuz Gölü brines, and Cenozoic marine brines in Jenecke units.

| Sample | Location, Well-Depth of Sampling, m | Janecke Unit, mol % (for Figure 7) | | | Janecke Unit, mol % (for Figures 8 and 11) | | | |
|---|---|---|---|---|---|---|---|---|
| | | 2K | Mg | SO₄ | Mg | 2Na | SO₄ | 2Cl |
| 14 | TG5-677.9 | 0.3 | 82.4 | 17.3 | 70.3 | 29.7 | 9.5 | 90.5 |
| 13 | TG5-889.8 | Fluid inclusions smaller than 40 μm | | | | | | |
| 12 | TG6-484.9 | 3.1 | 76.7 | 20.2 | 65.3 | 34.7 | 11.6 | 88.4 |
| 11 | TG6-618.4 | 3.2 | 60.6 | 36.2 | 45.5 | 54.5 | 23.0 | 77.0 |
| 10 | TG6-679.5 | Samples without primary fluid inclusions | | | | | | |
| 9 | TG6-728.5 | 3.7 | 82.8 | 13.5 | 75.1 | 24.9 | 7.6 | 92.4 |
| 8 | TG6-730.0 | 3.4 | 56.6 | 40.0 | 41.3 | 58.7 | 26.1 | 73.9 |
| 7 | TG6-762.7 | 2.0 | 83.0 | 15.0 | 73.7 | 26.3 | 8.3 | 91.7 |
| 6 | TG6-360.3 | 6.4 | 71.8 | 21.8 | 62.3 | 37.7 | 13.1 | 86.9 |
| 5 | TG7-422.0 | 3.4 | 76.0 | 20.6 | 64.7 | 35.3 | 11.9 | 88.1 |
| 4 | TG7-450.0 | 3.2 | 78.2 | 18.6 | 67.9 | 32.1 | 10.6 | 89.4 |
| 3 | TG7-455.0 | 1.9 | 70.3 | 27.8 | 60.9 | 39.1 | 14.0 | 86.0 |
| 2 | TG7-518.6 | 0.4 | 84.6 | 15.0 | 73.3 | 26.7 | 8.1 | 91.9 |
| 1 | TG7-562.9 | 4.4 | 83.6 | 12.1 | 77.7 | 22.3 | 6.7 | 93.3 |
| | Eocene seawater at the stage of halite crystallization, after [37] | | | | | | | |
| IX | Navarra, Spain (37 Ma) | 11.4 | 81.5 | 7.1 | 85.0 | 15.0 | 4.2 | 95.8 |

**Table 2.** *Cont.*

| Sample | Location, Well-Depth of Sampling, m | Janecke Unit, mol % (for Figure 7) | | | Janecke Unit, mol % (for Figures 8 and 11) | | | |
|---|---|---|---|---|---|---|---|---|
| | | 2K | Mg | SO$_4$ | Mg | 2Na | SO$_4$ | 2Cl |
| | Badenian seawater at the stage of halite crystallization, after [38] | | | | | | | |
| VIII | Carpathian area (13.8 Ma) | 8.8 | 75.7 | 15.5 | 71.0 | 29.0 | 9.3 | 90.7 |
| | Messinian seawater at the stage of halite crystallization, after [37,39–41] | | | | | | | |
| VII | Red Sea Basin (5.0–6.0 Ma) | 8.9 | 69.3 | 21.8 | 61.3 | 38.7 | 13.6 | 50.7 |
| VI | | 7.5 | 73.5 | 19.0 | 65.9 | 34.1 | 11.4 | 88.6 |
| V | Caltanissetta, Sicily (5.6–6.0 Ma) | 7.5 | 74.8 | 17.7 | 68.4 | 31.6 | 10.3 | 89.7 |
| IV | Lorca, Spain (7.6 Ma) | 8.7 | 69.3 | 22.0 | 61.1 | 38.9 | 13.7 | 86.3 |
| | Modern seawater at the stage of halite crystallization, after [35,36] | | | | | | | |
| III | Bahamian Islands (W46) | 6.6 | 69.0 | 24.4 | 58.7 | 41.3 | 15.1 | 84.9 |
| II | Bahamian Islands (W33) | 6.7 | 69.3 | 24.0 | 59.2 | 40.8 | 14.8 | 85.2 |
| I | Black Sea | 4.7 | 71.0 | 24.3 | 59.2 | 40.8 | 14.6 | 85.4 |

### 4.3. Sulfate Minerals in Rock Salt

The immersion studies of individual tiny crystals extracted from fluid inclusions showed that they are represented by gypsum, basanite, and anhydrite (Figure 5). Gypsum forms crystals ranging from a few microns to 100 μm with a prismatic or isometric outline and, sometimes, rounded corners (Figure 5A). A single crystal (or group of crystals) often fills almost the entire inclusion volume (Figure 5B). The bassanite (to 120 μm) forms acicular crystals, rarely with an elongated-prismatic habit (Figure 5C). Anhydrite occurs in tabular and elongated prismatic crystals (Figure 5D). It has also been found within halite crystals in the form of lamellar, lenticular, and elongated prismatic crystals in single halite chevrons and in the chevron bands.

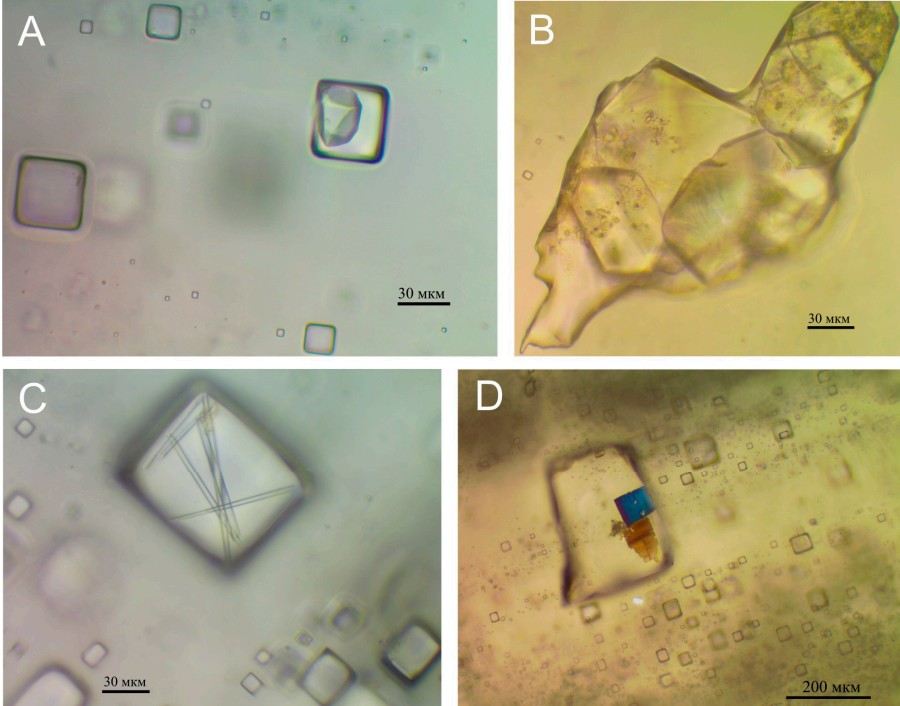

**Figure 5.** Calcium sulfate mineral crystals of fluid inclusions in halite. (**A**) Gypsum within primary inclusions, sample No. 11; (**B**) Gypsum in secondary inclusion, at the stage of diagenesis of rock salt, the dissolution of tiny gypsum crystals and the increase of individuals of this mineral took place in the newly formed large liquid inclusions, sample No. 4; (**C**) Bassanite, sample No. 11; (**D**) Anhydrite in secondary inclusion. For sample location, see Figure 2 and Tables 1 and 2.

The study of the water-insoluble residue of sedimentation halite by the XRD method showed that the insoluble residue from some samples (No. 2, 4, 14) is 100% represented by anhydrite. It is insufficient to perform an XRD analysis of the insoluble residue from samples 1, 5, 6, 7, 9, 11, 12, and 13. XRD analysis could not be performed in samples 1, 5, 6, 7, 9, 11, 12, and 13 due to insufficient insoluble residue. However, in the fluid inclusions of these samples, microcrystals of gypsum, basanite, and anhydrite were established via the immersion method, and the transparent halite of these samples contains microcrystals of anhydrite of various sizes.

Glauberite (Na$_2$Ca(SO$_4$)$_2$) is deposited either in sedimentary halite or in glauberite interlayers in rock salt of various samples. The glauberite crystals are colorless or yellowish beige (Figure 6) and have isometric, prismatic, and tabular habits. The color of the glauberite layers in rock salt range from greenish to grey and yellowish to white, depending on the amount of terrigenous material they contain.

According to XRD analysis, it was established that anhydrite is noted in the composition of glauberite monocrystals (sample No. 6 and No. 9 in the amount of 3% and 1%, respectively) and in the water-insoluble residue of sedimentary halite of sample No. 8 only glauberite is confirmed. In sample No. 3.97% glauberite, 3% thenardite, and less than 1% anhydrite is determined.

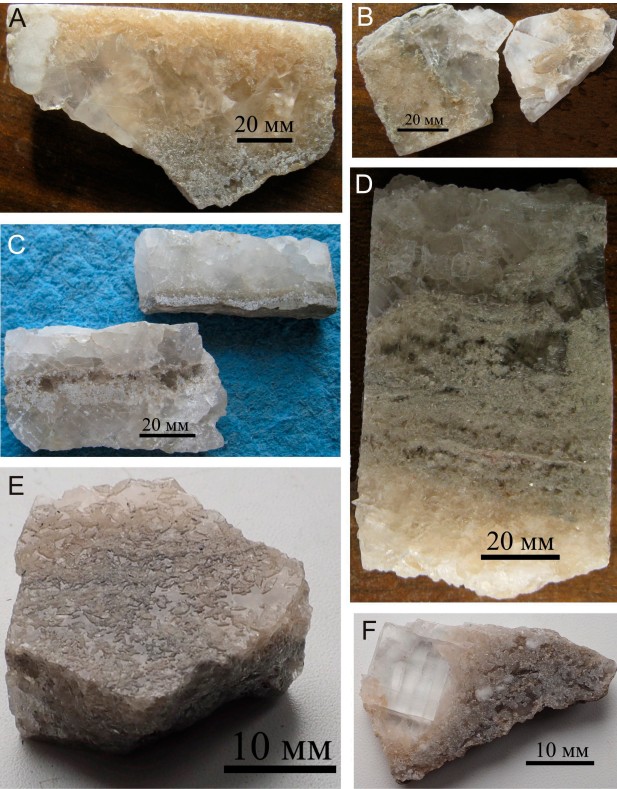

**Figure 6.** (**A**) Yellowish–beige glauberite in secondary halite, white-colored nodular anhydrite (left part), sample No. 10; (**B**) Sedimentary halite deposited after the formation of glauberite, sample No. 9; (**C**) A layer of fine-grained anhydrite in sedimentary halite, sample No. 4; (**D**) Sedimentary halite (in the upper part); greenish–grey clay-enriched interlayers with radial overgrowths of glauberite and tiny pores are filled with halite (in the central part), light glauberite layer in the form of isometric crystals and large pores and caverns are filled with halite (in the lower part), sample No. 12; (**E**) Pores and caverns in the glauberite layer are filled with secondary halite. Pores and caverns formed after finely dispersed gypsum or sodium syngenite dissolution, sample No. 12; (**F**) A layer of glauberite with residual white anhydrite and secondary halite. Glauberite growth occurred from solutions around the crystallization centers, represented by gypsum, sample No. 9. For Sample location, see Figure 2 and Tables 1 and 2.

### 4.4. Homogenization Temperature of Liquid Inclusions

Halite samples in the Tuz Gölü Basin contain single-phase (liquid) and two-phase (liquid+gas) inclusions. In addition, the gas phase was detected in the primary fluid inclusions of all samples, except for sample No. 11 (where it was found only in some large inclusions of more than 300 μm). This study has mainly examined the primary liquid–gas inclusions with a size of 40–80 μm located in a belt along the separated zones of sedimentary halite. Liquid inclusions larger than 300 μm were also studied in samples 11 and 8. Research has shown that a range of specific homogenization temperatures a characteristic of inclusions from different parts of chevron halite (Table 3).

**Table 3.** Homogenization temperature of liquid–gas inclusions in halite of well TG6. For sample location, see Tables 1 and 2.

| Sample | The Number of Investigated Bands of Inclusions from Different Growth Zones of Halite | Homogenization Temperature, °C | Crystallization Temperature, °C | Remarks |
|---|---|---|---|---|
| 12 | 4 | 14.5; 16.0; 16.8; 16.9; 18.5 15.7; 15.7; 16.6; 18.0; 18.6 14.3; 14.5; 17.8 13.3; 13.6; 14.0; 15.7 | 15.7–18.6 | The sample contains sedimentary textures with fluid inclusions of different phase compositions: 1. All fluid inclusions are gas–liquid 2. Gas–liquid fluid inclusions are only in certain zones |
| | 3 | 17.3; 17.4; 21.3; 21.5 19.5; 21.5; 21.9 21.3; 21.5; 22.0; 22.1 | 21.5–22.1 | |
| | 1 | 53.6; 55.4; 64.3; 68.7; 72.4 | 72.4 * | |
| 11 | 1 | 19.2; 20.0; 20.0; 20.6 | 20.6 | All sedimentary textures consist of single-phase fluid inclusions. Large gas–liquid inclusions occur in separate zones. |
| | 1 | 20.0; 20.3; 23.9; 24.0; 24.3 | 24.3 | |
| | 1 | 24.2; 27.5; 28.0; 28.6; 28.6 | 28.6 | |
| | 2 | 30.9; 33.6; 33.9 30.6; 33.3; 35.0 | 33.9–35.0 * | |
| 9 | 3 | 15.0; 15.1; 15.5; 16.7 14.1; 15.1; 15.5; 15.6 14.3; 14.5; 14.7; 17.6 | 15.6–17.6 | There are gas–liquid inclusions in the upper parts of sedimentary textures |
| | 1 | 18.6; 19.5; 21.0; 21.1 | 21.1 | |
| | 1 | 22.5; 23.7; 25.0 | 25.0 | |
| | 1 | 24.3; 25.0; 25.5; 25.5 | 25.5 | |
| | 1 | 40.2; 46.9; 47.9; 49.0; 49.1 | 49.1 * | |
| | 1 | 68.1; 69.9; 70.4; 72.0; 73.0 | 73.0 * | |
| 8 | 2 | 15.1; 17.3; 18.6 14.7; 15.0; 16.0; 17.0; 17.9 | 17.9–18.6 | The sample contains sedimentary textures with fluid inclusions of different phase compositions: 1. All fluid inclusions are gas–liquid 2. Gas–liquid fluid inclusions are only in certain zones 3. Large gas–liquid inclusions occur in separate zones 4. All fluid inclusions are single-phase |
| | 3 | 18.4; 18.9; 18.9; 19.1; 19.6 19.7; 20.6; 20.7 19.9; 20.4; 20.4; 21.6 | 19.6–21.6 | |
| | 2 | 21.0; 21.7; 23.1 20.1; 20.9; 23.0; 23.3 | 21.3–23.3 | |
| | 1 | 26.3; 26.5; 26.6; 27.0; 27.0 | 27.0 | |
| | 1 | 33.5; 33.5; 33.7; 33.9 | 33.9 * | |
| | 1 | 40.8; 41.0; 42.0 | 42.0 * | |
| | 2 | 59.4; 60.4; 61.8; 62.7; 62.9 53.8; 54.2; 56.8; 57.7; 61.8 | 61.8–62.9 * | |
| | 3 | 60.0; 65.1; 65.7; 66.2; 66.3; 71.0 59.0; 70.0; 73.0; 73.5 69.7; 70.4; 70.6; 71.7 | 71.7–73.5 * | |
| 7 | 1 | 12.7; 12.8; 14.0; 14.8; 16.6 | 16.6 | There are large gas–liquid fluid inclusions in the lower parts of sedimentation textures |
| | 2 | 17.4; 17.5; 18.1; 20.5 17.0; 19.3; 19.7; 19.9; 20.3; 20.3 | 20.3–20.5 | |
| | 1 | 39.2; 40.8; 40.8; 42.3; 44.1 | 44.1 * | |

\* High-temperature crystallization (conditions under which fluid inclusions with a gas phase are formed).

Individual gas–liquid inclusions are mostly large and homogenize at a temperature of 95–115 °C and above or do not homogenize.

*4.5. Bromine Content in Halite*

Bromine content in halite from TG5, TG6, and TG7 boreholes ranges from 2.92 to 47.17 ppm, and the average value is 15 ppm (Table 4).

**Table 4.** Bromine content in halite in boreholes in the Tuz Gölü Basin.

| Sample | Well Depth of Sampling, m | Br Content, ppm |
| --- | --- | --- |
| 14 | TG5-677.9 | 47.17 |
| 11 | TG6-618.4 | 19.00 |
| 9 | TG6-728.5 | 36.00 |
| 8 | TG6-730.0 | 15.07 |
| 7 | TG6-762.7 | 6.15 |
| 5 | TG7-422.0 | 15.10 |
| 4 | TG7-450.0 | 6.32 |
| 3 | TG7-455.0 | 2.92 |
| 2 | TG7-518.6 | 8.86 |

## 5. Interpretation and Discussion

*5.1. The Chemical Composition of Messinian Sedimentary Brines from the Tuz Gölü Basin*

The formation mechanism of halite sedimentary structures has previously been studied in detail in modern salt lakes [36,42]. It was found that the sedimentary textures of halite identified in ancient salt deposits contain fluid inclusions of various origins [28,43]. Additionally, the chemical composition of brines in primary fluid inclusions in different bands of sedimentation textures may differ slightly [3]. Thus, in our study, groups of inclusions from a single band of chevron halite were examined for each sample.

According to the ultramicrochemical analysis data, the Tuz Gölü brine is of the sulfate type (magnesium sulfate subtype, see Table 1), similar to the Messinian marine brines [39–41].

The obtained $r$Na/$r$CI values for the Tuz Gölü range from 0.80 to 0.97, indicating the continental–marine origin of the brine. This sodium-to-chlorine ratio for seawater in the halite deposition stage is always below 0.90 [35], and this ratio in continental sodium sulfate waters is greater than 1.

In modern continental sea lakes containing seawater relics, depending on the volume of sea water in them and the nature of the continental runoff, the $r$Na/$r$CI ratio increased from 0.87 (seawater) to 0.93 (Caspian Sea) and 0.99 (Aral Sea) [44]. The continental–marine origin of the Tuz Gölü brine confirms the previously obtained isotope data [5,7,8], which indicates recurrent marine transgressions in the study area.

In mineral paragenesis, continental evaporites are much more diverse than marine ones. However, in many cases, the difference between the two evaporites is only conditional, which causes some difficulties in reproducing the paleohydrological conditions and water–salt regime (balance) of salt basins [45]. The mineral composition of the Tuz Gölü rocks varies from glauberite and halite to gypsum/anhydrite, less frequently magnesite. Glauberite and gypsum/anhydrite occur more commonly in the study area. During isothermal processes (at 25 °C), from marine Na–K–Mg–Cl–SO$_4$ seawater, only calcite, gypsum, halite, epsomite, kainite, carnallite, and bischofite are formed, while glauberite, mirabilite, thenardite, astrakanite are not precipitated [46].

Compared to the Na–K–Mg–Cl–SO$_4$ (SO$_4$-rich) type marine brines of the stage of halite deposition, the Tuz Gölü brine has slightly higher [Na$^+$] and lower [K$^+$] concentrations. Based on the [Mg$^{2+}$] and [K$^+$] data, it was determined that the basin brine concentration changed twice during salt accumulation. The concentration of [SO$_4^{2-}$] varies over a wider range than other ions. Similar to the concentration of [Mg$^{2+}$], it sometimes reaches values typical of marine brines at the beginning of halite deposition [35]. The sharp decrease in [K$^+$] concentration in the brine, recorded at certain stages, is related to the interaction of brines with organic matter and clay of continental origin, which enters the basin with wind and continental runoff. It is known from the literature that in the brines of marine evaporite

basins surrounded by land or lakes separated from the sea, the sodium content decreases slightly, and potassium may disappear completely [34–36,46–50].

In salt basins of marine origin, if potassium salts are present during halite precipitation, the ratios between major ions in brine remain stable [51]. Some decrease in the concentration of $[SO_4^{2-}]$ in Na–K–Mg–Cl–$SO_4$ ($SO_4$-rich) type marine brines may be due to the influx of continental waters enriched in $Ca(HCO_3)_2$ into the saline basin. However, with a significant volume of seawater, such an impact does not significantly change the ratio of alkali ions in the basin's brines [52]. In continental basins with sulfate-type halogenesis, during halite crystallization and before the onset of astrakanite deposition, brines are also characterized by the stability of the ratio between major ions [36].

A comparison of the brine of the Tuz Gölü Basin with similar marine brines in the Eocene, Miocene (Badenian and Messinian), and modern period are presented by plotting the points of their chemical composition on the Mg–2K–$SO_4$ ternary diagram (Figure 7).

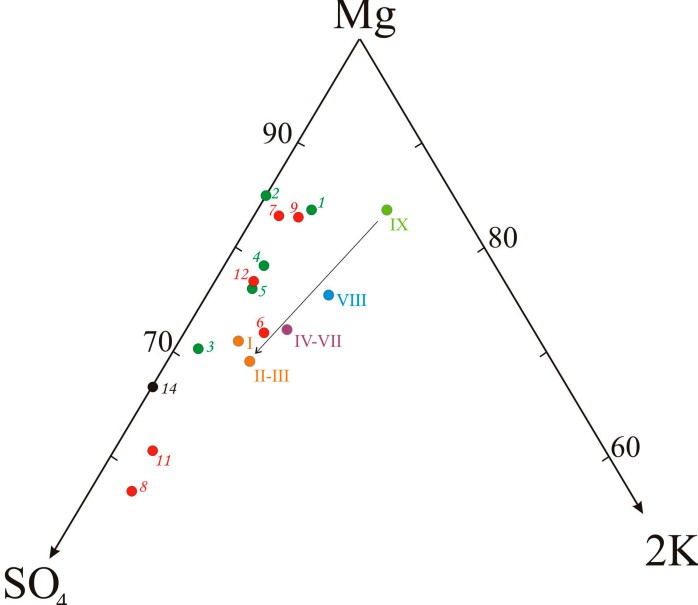

**Figure 7.** The chemical composition of brines on the Mg–2K–$SO_4$ chart; point numbers according to Table 2.

In the diagram (Figure 7), the distribution of points of seawater composition in the Cenozoic is mainly related to the evolution of seawater composition over time (shown with an arrow). The distribution of points of the Tuz Gölü Basin is due to the influence of different salt sources and physical–chemical processes in the salt basin, mainly reflected in the fluctuations of $[SO_4^{2-}]$ concentration in the brines. Sources of sulfate ions in the Tuz Gölü Basin were continental leaching of older Oligocene and Eocene evaporites as well as seawater [7] and products of Upper Miocene volcanism in the western part of the basin [53]. In such waters, which drain sulfate-facies rocks, the main constituents are calcium and sodium sulfates and, to a lesser extent, calcium and magnesium bicarbonates [44].

In the $Na_2Cl_2$–$MgCl_2$–$MgSO_4$–$Na_2SO_4$ diagram (Figure 8), the halite crystallization line's angle depends on the brine's sulfate content. The sharpest angle from $Na_2Cl_2$–$MgCl_2$ forms the crystallization line of Eocene marine halite, the least sharp—of modern marine halite, following the progressive increase in the content of sulfate ions in seawater during the Cenozoic period. The crystallization lines, which are significantly changed by continental factors (Garabogazköl, Kirleut Lake), are distant from the lines of normal seawater crystallization. The Garabogazköl brines are enriched with continental sulfate, while the ones in Kirleut Lake are poor. In the $Na_2Cl_2$–$MgCl_2$–$MgSO_4$–$Na_2SO_4$ diagram, the Tuz Gölü halite crystallization line does not coincide with the Messinian marine halite crystallization line (IV–VIII) (see Figure 8). However, it is located close to it and also close

to the crystallization lines of Cenozoic seawater halite (I–IX). The shift of points downward from the halite crystallization line is related to the formation of glauberite during periods of suspension of halite precipitation and the removal of sulfate and some sodium ions from the brine composition. This process is confirmed by the obtained data on the chemical composition of the parent brines of the halite deposition stage (see Section 5.2.2). As a result of this process, "false marine brines" with a $MgSO_4/MgCl_2$ ratio close to that of sea brine were formed (see Table 1).

The samples we studied share characteristics with the glauberite–halite lithofacies (A) described for the Zaragoza Gypsum Formation (Lower Miocene, Ebro Basin, NE, Spain) [54]. The particularity of the chemical composition of brines in which halite precipitated in the Ebro Basin is explained by the depletion of sulfates in the heavy brines as a result of glauberite crystallization [54].

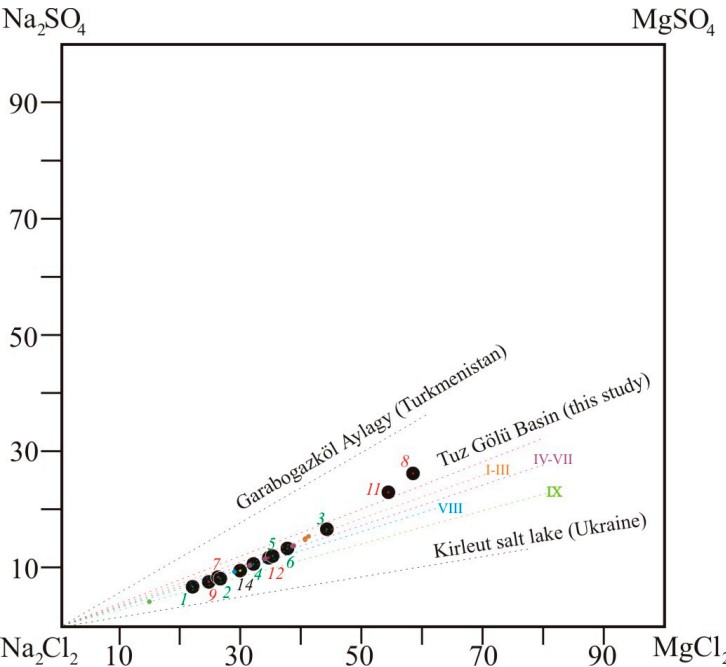

**Figure 8.** Chemical composition of brines in the $Na_2Cl_2$–$MgCl_2$–$MgSO_4$–$Na_2SO_4$ diagram, point numbers according to Table 2. The straight lines passing through the $Na_2Cl_2$ vertex and forming an acute angle with the $Na_2Cl_2$–$MgCl_2$ line are individual due to the crystallization of halite in each basin [55].

### 5.2. Peculiarities of the Formation of Sulfate Minerals in the Studied Samples

#### 5.2.1. Calcium Sulfate Minerals

Gypsum is the only mineral form of calcium sulfate that crystallizes simultaneously with halite during brine evaporation [56]. In the saline Tuz Gölü Basin, tiny gypsum crystals nucleated during the influx of "fresh" waters with $Ca(HCO_3)_2$ and were deposited on the bottom of the basin. They were captured on the surface of the halite crystals oriented parallel or sub-parallel to the bottom of the basin (Figure 9). In the studied fluid inclusions, the liquid inclusions include bassanite, anhydrite, and gypsum. Tiny gypsum crystals have been preserved in those inclusions, occupying more than 5% of the inclusion volume. The transformation of gypsum–bassanite–anhydrite and the growth of gypsum crystals in large inclusions of diagenetic halite followed a similar pattern described earlier [57,58].

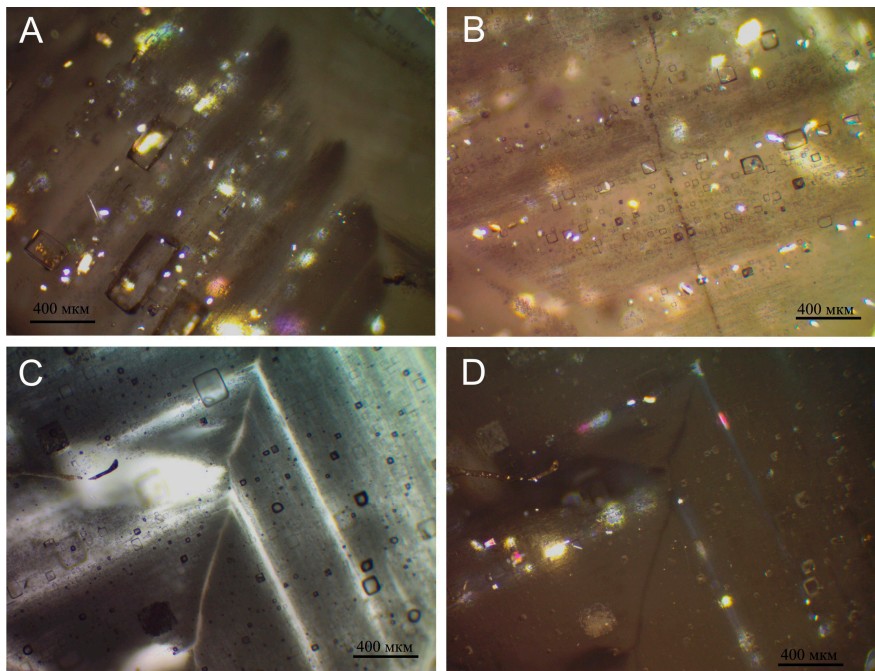

**Figure 9.** Gypsum nucleation during the formation of bottom-grown halite. (**A**,**B**) Nucleation during the day–night period; sample No. 6 (**A**); sample No. 1 (**B**); XPL crossed polarized light. (**C**,**D**) Nucleation mainly during the night; sample No. 7 (**C**) PPL plane polarized light; sample No. 7 (**D**), XPL crossed polarized light.

### 5.2.2. Sodium Calcium Sulfate

Glauberite crystals concentrated in glauberite layers were formed when halite deposition was suspended (Figure 6). The formation of glauberite occurred as a result of the slow dissolution of previously deposited finely dispersed metastable minerals—gypsum ($CaSO_4 \times 2H_2O$), sodium syngenite ($Na_2SO_4 \times CaSO_4 \times 2H_2O$), or mirabilite ($Na_2SO_4 \times 10H_2O$) [59,60].

Evidence of such formation is the deposition of anhydrite in glauberite monocrystals and residual anhydrite within glauberite layers (see Figure 6). After most of the gypsum was dissolved, glauberite was formed. Pores and caverns remained filled with halite (see Figure 4). The decrease in the sulfate ion content in Na–K–Mg–Cl–SO$_4$ brines of ancient salt-bearing basins is correlated with the intensity of crystallization of sulfate minerals when continental waters containing Ca(HCO$_3$) enter the basins [61]. Obviously, the processes of glauberite formation during the suspension of halite precipitation caused the sulfate ion to be removed from the composition of the brines, and the reduced concentration of this ion in the brines led to the reprecipitation of halite (see Table 1).

Under certain physicochemical conditions, glauberite crystallization also coincided with halite sedimentation (Figure 10). The area of its crystallization in the diagram (Figure 11) has not been precisely defined. The lower limit of the glauberite formation area on the diagram has been clarified in the study conducted. In this area of the diagram, glauberite was formed under the influence of magnesium sulfate brines enriched in sodium during the dissolution of gypsum in two stages: first, finely dispersed gypsum (1) crystallizes, when interacting with sodium brine and then turns into a metastable form of glauberite ($2Na_2SO_4 \times CaSO_4 \times 2H_2O$). A similar process is typical in modern salt lakes [62]

$$Ca(HCO_3)_2 + MgSO_4 \leftrightarrow CaSO_4\downarrow + Mg(HCO_3)_2\downarrow$$

$$CaSO_4 + 2H_2O \quad xMg(OH)_2 \times yMgCO_3$$

(1)

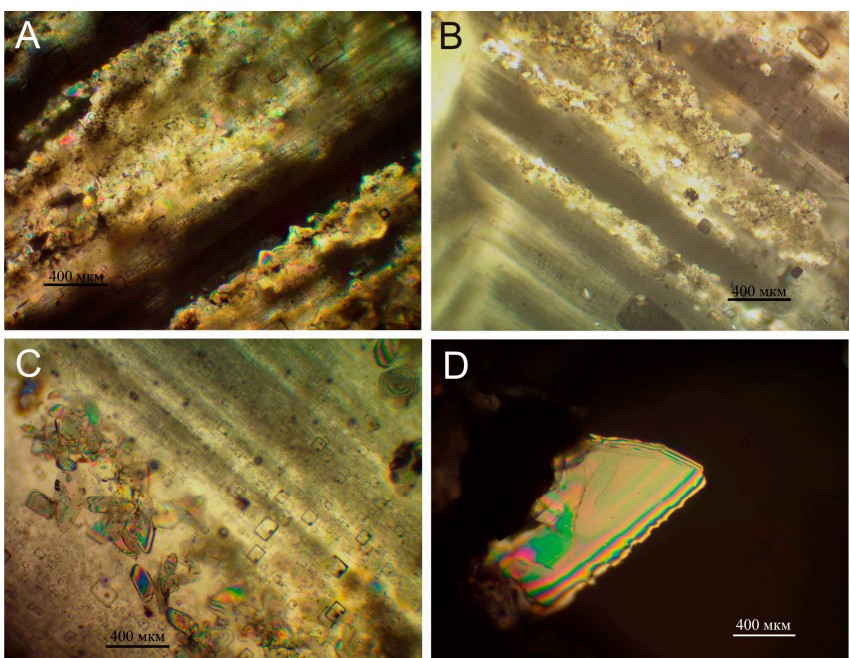

**Figure 10.** Glauberite crystals in chevron halite. (**A**,**B**) Sample No. 3, XPL; (**C**) Sample No. 8, XPL. (**D**) Partial dissolution of the crystal of the glauberite layer during the renewal of the halite sedimentation, sample No. 9, XPL.

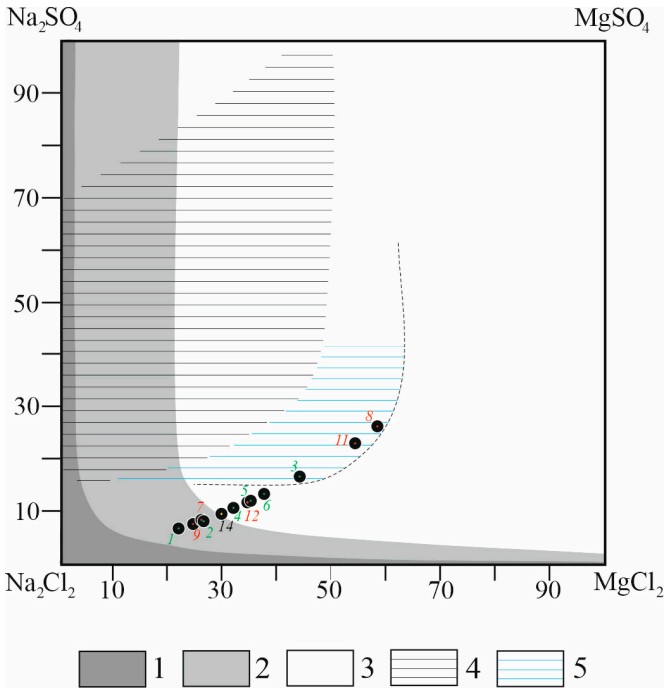

**Figure 11.** Diagram of $Na_2Cl_2$–$MgCl_2$–$MgSO_4$–$Na_2SO_4$ with crystallization fields of solid phases, which are products of the reaction of brines of the sulfate system with calcium bicarbonate [62]. Crystallization fields: 1—calcite, 2—calcite, gypsum, and basic magnesium carbonates, 3—gypsum and basic magnesium carbonates, 4—glauberite according to [62], 5—glauberite (our new data). Point numbers according to the Table 2.

During the precipitation of halite at a reduced concentration of $[SO_4^{2-}]$ (samples No. 1, 2, 4–7, 9, 12, 14), newly formed gypsum both in zonal halite and in thin layers with terrigenous material did not transform into glauberite (see Table 3, Figure 6). However, a high sulfate ion content was also insufficient for such a transformation (for example, glauberite is absent in sample No. 11).

5.2.3. Sodium Sulfate

Mirabilite in the sediments was not found, but according to XRD analysis, the presence of thenardite was noted in the insoluble residue of individual samples (see Section 4.3). The simultaneous deposition of thenardite and halite in sedimentation basins does not occur, but only the simultaneous deposition of mirabilite and halite is possible [55]. In modern sulfate lakes with a high concentration of NaCl, the precipitation of mirabilite is observed in the summer on some cool nights when the brine temperature drops below 5 °C. When the temperature increase to 15.3–16.0 °C, mirabilite turns into thenardite [55,63,64]. Halite in the Tuz Gölü Basin crystallized at temperatures exceeding 15 °C (see Table 3). Whether the temperature of the brines dropped to 5 °C overnight requires additional research. For example, in the range of crystallization temperatures of Permian halite in North America, which was 21–50 °C, the daily (day–night) crystallization temperature ranged from 6 °C to 26 °C [65].

*5.3. Paleoclimatic Characteristics*

Single-phase fluid inclusions, which we observed in halite at room temperature, are characteristic of crystals grown at temperatures lower than 50 °C [28,37,66,67]. Such inclusions are typical for the halite of modern salt lakes and many ancient salt deposits [4,33,68,69].

The lower value of the crystallization temperature when the gas phase occurs in fluid inclusions in halite is not precisely established. Its appearance depends on the size of the inclusions, chemical composition, and concentration of brines in the inclusions. At the crystallization temperature of the Tuz Gölü halite exceeding 42.0 °C, the gas phase appeared in most fluid inclusions, at the crystallization temperature of 33.9–35.0 °C—only in some large-sized inclusions. At a lower crystallization temperature, the gas phase in inclusions did not occur. Individual large gas–liquid inclusions are not homogenized, or their homogenization occurs at 95–115 °C, which is associated with partial or complete depressurization of these inclusions along stress lines in halite during the research preparations (Figure 12).

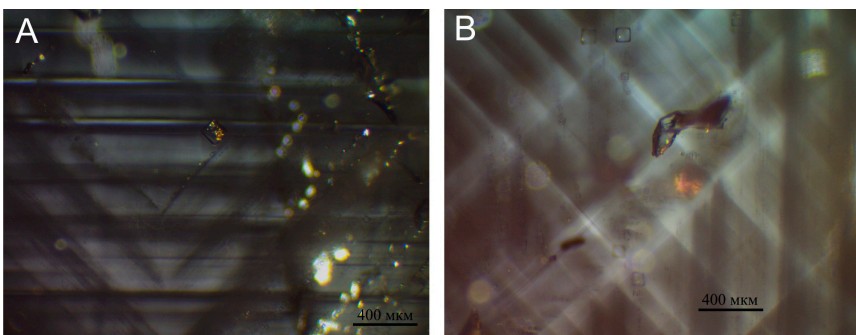

**Figure 12.** (**A**,**B**) Stress lines in halite, XPL, sample No. 11.

Homogenization temperatures of inclusions were obtained at specific intervals in the Tuz Gölü halite (Table 3). The actual crystallization temperature of the mineral corresponds to the maximum specified value of the homogenization temperature [66,70]. Since we determined the temperature of the bottom layers of water, the obtained data indicate the depth of the sedimentation basin, in which the entire brine layer responded to changes in air temperature. The set temperatures characterize its daily changes [65]. They were 15.6–49.1 °C during summer crystallization and 15.7–22.1 °C in spring or the cold summer–autumn period. At the end of spring and the beginning of summer, a large amount of

conifer pollen and spores of other terrestrial plants entered the basin. The daytime air temperature fluctuated between 20.6 and 35.0 °C (see Table 3; Figure 4).

During some periods, the Tuz Gölü halite crystallized at 61.8–73.5 °C. The extreme high-temperature crystallization regime at the bottom of the salt basin was achieved by the emergence of vertical thermohaline zoning in the sedimentary basin with an upper up to 15 cm fresh or weakly saline transparent water layer—mixolimnion (formed to the inflow of surface/ground freshwater, precipitation), lower—chemocline, and bottom—monimolimnion [71]. The sun's light energy in the chemocline was converted into thermal energy and used to heat the mixolimnion. The inversion thermohaline zoning of the Tuz Gölü waters was temporary. Its destruction occurred as the mineralization of the mixolimnion water increased until it equaled or exceeded the mineralization of the mixolimnion. According to research, although the "greenhouse effect" in the Tuz Gölü was established only for a short period, it was periodically renewed due to the influx of "fresh" waters. Similar phenomena of stratification of the water column and establishment of high bottom temperatures are recorded in some ancient marine salt-bearing basins that precipitate halite [72,73]. The maximum temperatures of bottom brine, measured in modern natural lakes of the world with thermohaline zoning, are 70 °C [74].

The obtained homogenization temperature of the inclusions in the halite directly correlates with data from isotopic studies (C, O, Mg) [5], (Sr, S, O) [6,7], and (B, Br, Cl, and Li) [8], indicating rapid temperature fluctuations during deposition and burial of the evaporites [5,7].

### 5.4. Peculiarities of Bromine Content in Tuz Gölü Halite

At the beginning of marine halite precipitation, it contains 68 ppm of bromine [75], while it contains 160 ppm before the crystallization of potassium–magnesium salts [76]. Halite having lacustrine origin usually contains 1–10 ppm bromine [77] but can reach 180 ppm, for example, in Holocene–Pleistocene halite from non-marine Searles lake in California [78].

The bromine content in the Tuz Gölü halite in TG5, TG6, and TG7 boreholes ranges from 3 to 47 ppm (see Table 4). The bromine content in the halite from other intervals of the TG6 borehole is close to these values and is between 5 and 19 ppm (average 12 ppm) [8]. In the borehole conducted approximately 2 km northwest of the TG5 borehole (TG 4), it was determined that the Br contents of the pure and nearly pure halite range from 18 to 637 ppm (average 234 ppm) [9]. Therefore, we can assume a significant role in forming salts of continental and bromine-depleted deep waters. However, repeated redeposition of marine halite also causes a decrease in its bromine content to 10–15 ppm and below [75,79–81]. The abnormally high values of the bromine content in halite are due to the high concentration of brines in some regions of the basin or the inflow of bromine-enriched deep waters, which was possible in the case of significant differentiation of the paleotectonic conditions of sedimentation. The increased concentration of the Tuz Gölü brines in some areas can be indicated by the findings of astrakanite, epsomite, loweite, and starkeite in saline deposits, along with the most common evaporite minerals of the basin, which are halite, glauberite, anhydrite, gypsum, dolomite, and magnesite [8]. The question of the specifics of the distribution of bromine content in halite over the basin area requires additional study.

### 6. Conclusions

1.  Using an ultramicrochemical method, it was determined that the halogenesis in the Tuz Gölü Basin in Messinian corresponds to the sulfate type and the magnesium sulfate subtype. Compared to the Na–K–Mg–Cl–$SO_4$ marine brines from the halite deposition stage, the Tuz Gölü has slightly higher [$Na^+$] and lower [$K^+$] concentrations. The basin brine salinity during salt accumulation changed twice based on the [$Mg^{2+}$] and [$K^+$] concentration data. The concentration of [$SO_4^{2-}$] varies over a wider range than other ions. Similar to the concentration of [$Mg^{2+}$], it sometimes reaches typical values of sea brines at the beginning of halite precipitation. A significant decrease in

the concentration of $[SO_4{}^{2-}]$ in sedimentation brines from 18.2 to 4.5 g/L is caused by physicochemical processes in the near-surface and bottom layers of the basin water during the suspension of halite precipitation. During these periods, an intensive inflow of $Ca(HCO_3)_2$ into the sedimentary basin occurred, and glauberite layers formed. Under certain physicochemical conditions, crystallization of glauberite from newly deposited finely dispersed gypsum co-occurred with halite sedimentation.

2. It was established via XRD and immersion methods that within fluid inclusions in halite, allogeneic calcium sulfate crystals are represented by gypsum, bassanite, and anhydrite. Sulfate minerals in halite crystals are glauberite, anhydrite, and thenardite. During halite deposition at reduced concentrations of $[SO_4{}^{2-}]$, the newly formed gypsum did not transform into glauberite in zoned halite and thin layers with terrigenous material. However, the high content of sulfate ions was also not a sufficient condition for such a transformation.

3. By thermometric studies, fluctuations in daily and seasonal air temperatures in the Tuz Gölü basin were determined. Climatic indicators during the Messinian halite forming/precipitation in the region revealed rapid changes in daytime air temperature from 15.7–22.1 °C in spring or cool summer–autumn, 20.6–35.0 °C in late spring–early summer, and up to 15.6–49.1 °C in summer. During some periods, Tuz Gölü halite crystallized at extremely high temperatures of 61.8–73.5 °C. The "greenhouse effect" in the basin was created for a short time but periodically renewed due to the influx of "fresh" waters.

4. The study of halite homogenization temperatures indicates that the depth of the basin was shallow during sedimentation, and the study of bromine in halite reveals the variation of the basin's paleotectonic conditions and the partial or temporary separation of different sections of the broad basin from each other.

**Author Contributions:** Conceptualization, A.R.G., M.Ç.K., and Y.Y.; methodology, A.R.G. and Y.Y.; investigation, M.Ç.K., and N.K.; resources, M.Ç.K. and N.K.; writing—original draft preparation, A.R.G. and K.B; writing—review and editing, K.B., A.R.G., and M.Ç.K.; supervision, A.R.G. and M.Ç.K.; project administration, A.R.G.; funding acquisition, M.Ç.K. and K.B. All authors have read and agreed to the published version of the manuscript.

**Funding:** This research was funded by TÜBİTAK, grant number 114Y629, and preparation of the research results by grant AGH number 16.16.140.315 (KB).

**Acknowledgments:** This research was made possible with the support of The Scientific and Technological Research Council of Türkiye (TUBITAK 114Y629). We also thank MTA for helping us to acquire the drilling samples.

**Conflicts of Interest:** The authors declare no conflict of interest.

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
