# Peer review of "Geochemistry of Brine and Paleoclimate Reconstruction during Sedimentation of Messinian Salt in the Tuz Gölü Basin (Türkiye): Insights from the Study of Fluid Inclusions"

_minerals, doi:10.3390/min13020171_

Round 1

Reviewer 1 Report

I’ve reported on the manuscript “Geochemistry of Brine and Paleoclimate Reconstruction During Sedimentation of Messinian Salt in the Tuz Gölü Basin (Türkiye): Insights from the Study of Fluid Inclusions” dealing with the reconstruction of the physical parameters responsible for the formation of salt-rich deposits in a lake systems affected by periodic fresh water inflows. The petrofacies (halite, glauberite and Ca-sulfates) and the geochemical data discussed in the ms are potentially very interesting for the evaporitic community and , more in general, the geological community; however, there are several points that can be implemented or modified in order to make the paper more readable. Some method should be better explained in the main text; this is especially evident for methods briefly cited using references written in Russian and or Ukrainian. English must be checked carefully, at least from a not-native speaker.

Below I’ve provided point-by-point minor comments that I hope will improve the quality of the manuscript.

Given these modifications, I’m in support of publication for this research.

Point-by-point comments:

Line 3 (title):  Here and throughout the text, since this is an international Journal, I suggest to use the English form: Turkey instead Turkiye.

Line 18: bottom water layers, change with “ bottom waters”

Lines 24-25: I suggest rewording of this sentence: “The sulfate minerals ….. are found within halite crystals.”

Line 27. Not clear the reason to report such 3 ranges of temperature in the abstract. It should be clarified or changed with a single range.

Line 59. Emanations: maybe better to change with “emissions”?

Line 61. Isotopic studies: please specify which type and include the range of temperature fluctuations.

Line 67. “Objectives”, three main objectives are reported.

Line 67. Delete “based on the obtained data”.

Line 70. “Depend” change with “depending on”

Line 99. I suggest to change “Then, due to” with “As a consequence of..”

Line 101. During the MSC… instead of “Concerning”…

Line 102. “They” is it referred to Akgun et ?

Lines 117-119. I suggest rewording: The Pliocene-quaternary succession consists of…. clay-rich deposits ()….

Fig. 1. Konya basins. This name is not specified in the main text.

Lines 124-127. These information are more adequate for the Material and Method section including coordinates of the studied wells.

Fig.2. In the legend of the map, the Paleocene-Eocene rocks are indicated as sedimentary rocks, other older rock are described with the name of the Formation. Please be consistent and make clear which type of sed. Rocks. Moreover, I suggest to change “Discordant” with “Unconformity”. In the well logs, a very general Miocene-Pliocene age is indicated; is there any more accurate time constrain allowing, for instance, for the identification of the Mio-Pliocene boundary?

Line 133. Ultramicrochemical method. I think that this method should be briefly described to the readers of the Journal and not only referred to a specific reference.

Line 139. Is it the concentration of the ions that decrease after 2-3 analyses? Please clarify

Line 141. Bunsen method: please add a reference.

Line 148. Immersion method. Again, this method should be better explained in this section; the reference provided here (and for many other citations in the manuscript) is in Russian and not readable for most of the readers.

Line 161. Which are the main characteristics of this chamber?

Line 169. “The size of the cumulate halite crystals reach…”

Fig. 3. In B, is the portion between the v-shaped chevrons a dissolution pipe? I would describe these feature in the text. Moreover, I suggest using letters or arrows to indicate the main features in the figures (e.g. dark fluid inclusion-rich bands,…)

Line 177. I suggest to change “zones” with “bands” and “rhythm” with “cyclic pattern”.

Line178. Why only 13 cycles have been distinquished? Is related to the dimension of the sample and this is the largest sample analyzed? Or are there dissolution surfaces that interrupt the sedimentation?

Line 179. Change “zone” with “band”

Line 180. Axis: which axis? Please clarify also in the figure.

Line 182. This sentence is not clear. Does it mean that the salting-out halite has not been recognized in the studied samples?

Fig. 4. Please add some more detail and be more specific in the caption of these nice figures.

Line 203. “some periods”, please change with “some samples”.

Line 211. Change “and amount to” with “with amount of”

Line 214. Not clear which is mentioned study here.

Table. Please clarify the two numbers in the brackets (ie. sum of single ions). Sycilly: change with Sicily

Line 221-223. Not clear how the percentages of the individual ions were calculated. Please clarify

Table 2. “Janecke unit” not clear why it is indicated in the topmost line. Not present in Tab.1

Line 228. Columnar and isometric: not clear this habit of the crystals. Are they prismatic and tabular or elongated?

Line 234. Between zones of chevron: are these dissolution pipes?

Fig. 5. Very nice figures, but please add some arrows and/or letter to help the reader with the caption.

Line 249-250. Not clear how glauberite is disposed in the samples. In this sentence I would rather state that glauberite is found either as isolated crystals within the bottom-grown halite or in distinct layers…. Please cite here Fig. 6

Lines 254-257. I would rather reword like: “According to XRD analyses, anhydrite was detected within glauberite monocrystlas ….. “ Is it the correct meaning of this sentence?

Line 257. Installed: ???

Fig. 6. See my previous comment on the other captions and figure (eg, labelling).

Line 273 and following. In the method section, it was stated that the vapour (gas) phase was induced artificially by a low cooling. Here it is not clear if the described and anlyzed FI are two-phases already before the artificial cooling. This is rather important and could explain some “strange” homogenization temperature; if this is case (FI L+V in origin), I don’t think that the microthemometric data are realiable. Please explain

Table 3. Please explain the meaning of crystallization temperature and how is it calculated? Why is there a range of temperatures? Moreover, the column relative to the investigated band indicates the number of bands (eg dark) in which FI data have been obtained? For example, 1 means that homegen. Temperatures derive from a single band? In the remark column should be indicated if the measured inclusions were 2 phase or monophase in origin. Are the large gas-liquid inclusion reliable or they suffered some diagenetic overprint?

Line 291. Why “sedimentary”? I suggest deleting

Line 296. This is confusing. I suggest to change”different zones of sedimentation textures” with in different halite lithofacies and in different portion of an individual lithofacies”

Line 297. See my comment in the table 3: be more specific on the site of analysis of the fluid inclusions (single band, a group of bands,.. )

Line 299. … as oberved in Messinian marine brines elsewhere (add citation)

Line 321-322. Data: change with “concentrations in the studied salt successions, the composition of the basin brine…”

Lines 324-325. It is not clear myself the relations between K decrease and the brine interaction with organics and/or clay mineral. Please explain

Line 341. Change “triangular” with “ternary”

Fig. 7. This figure need a readable legend with description of color coding and samples ID (type of samples and provenance)

Line 351. Dissolved onshore, process commonly described as continental leaching of older evaporites

Line 367-369. Which type of glauberite is forming according to this process? I suggest to include in the discussion (here or elsewhere) a comparison with the lithofacies described by Salvany et al 2007 in the Ebro Basin.

Line 385. Bassanite

Lines 387-389. There are continuous citations to interesting processes, methods and petrofacies described in articles written in Russian or Ukrainian. I think that some more detail of these process should be reported in the text given that Minerals is an international journal and the official language is English.

Fig. 9. In the caption: I suggest to change sedimentary halite with bottom-grown halite, here and throughout the text.

Lines 391-392. Day-night period: this is an interpretation not a description of the figure; better: alternation of clear and dark bands.

Lines 395-399 and following. Consider to compare the glauberite lithofacies with those of Salvany et al 2007 (Sedimentology)

Lines 447-448. not clear why is provided here an example of the Permian halite. I would conclude with something like: further analyses are needed to check the T drop overnight..

Lines 455-464. Not clear this paragraph; I’m not sure about the meaning of the appearance of the gas bubble. Is it related to process of leaching, collapse of the inclusions walls..? Please explain. It seems that many inclusions have varied their initial volume (e.g presence of large vapour bubble) affecting the crystallization temperature.

Line 463. Delete ”preparation of”

Lines 470-473. I do not agree. The obtained data are not indicative of shallow water, these mast be supporte da facies analyses.

Lines 473-473. How the Authors have recognized the seasonality (spring-summer)? Is there any link with the pollen remains whitin fluid inclusion?

Line 481. 15 cm of fresh water: which is the origin of this data? Is it theoretical or it is related to Tux Golu?

Line 494. Which type of isotopic studies?

Lines 547 and following. This point is not clear. Why the basin was very shallow? For instance in the the Dead Sea the temperature of halite nucleation ranges from 24° to 36° (see Sirota et al 2016 and following papers).

Author Response

Response to Reviewer 1 Comments

Thank you for reviewing.

The remarks posted were very helpful to us and helped us improve the text and avoid errors. We especially thank you for your many critical comments; we appreciate the work, and the vast majority of the corrections have been incorporated into the text (the most important modifications and changes are highlighted in color). In several cases, we have written appropriate explanations for this review (below).

SOME EXPLANATIONS

Line 3 (title): Here and throughout the text, since this is an international Journal, I suggest to use the English form: Turkey instead Turkiye

Explanation: The name Turkey was approved by the United Nations and changed to Türkiye. Türkiye will now be used instead of Turkey. I added the confirmation link: https://turkiye.un.org/en/184798-turkeys-name-changed-turkiye

Line 59. Emanations: maybe better to change with “emissions”?

Explanations: emenations is appropriate term for volcanism.

Line 61. Isotopic studies: please specify which type and include the range of temperature fluctuations.

Explanation: Isotope analyzes of various elements made in evaporite minerals were written with reference. The evaluation of the formation temperatures of the minerals from these isotope results was not written.

Fig 1: Konya basin

Explanation:

I modified the first sentence of the Geological Setting to include the Konya closed basin as:

“The Tuz Gölü Ð’asin is located in the south of Konya closed basin and covers some sub-basins (Haymana-Bala and EreÄŸli-Ulukışla). The basin is the largest (62 000 km2) inner basin among the Central Anatolian Cenozoic basins (Çankırı-Çorum, Yozgat, and Sivas).”

Fig.2. In the legend of the map, the Paleocene-Eocene rocks are indicated as sedimentary rocks, other older rock are described with the name of the Formation. Please be consistent and make clear which type of sed. Rocks. Moreover, I suggest to change “Discordant” with “Unconformity”. In the well logs, a very general Miocene-Pliocene age is indicated; is there any more accurate time constrain allowing, for instance, for the identification of the Mio-Pliocene boundary?

Explanation: All of the suggestion were made on map. Miocene-Pliocene units are compatible with each other. Generally, Volcanism was started in the Upper Miocene, but continued until the Pliocene-Quaternary and the young volcanism defined as Hasandağı volcanism. There is no clear boundary between the Miocene and Pliocene in the outcrop and boreholes.

Lines 124-127. These information are more adequate for the Material and Method section including coordinates of the studied wells.

Explanation: Location of TG-5, TG-6 and TG-7 boreholes are shown on the map (Figure 2), and also Figure 2 include the coordinate data.

Line178. Why only 13 cycles have been distinquished? Is related to the dimension of the sample and this is the largest sample analyzed? Or are there dissolution surfaces that interrupt the sedimentation?

Explanation: Among the samples studied, 13 rhythms (cyclic pattern) are the most significant. Among the factors that influenced this, the orientation of the crystal at the bottom of the basin was a decisive factor.

Line 182. This sentence is not clear. Does it mean that the salting-out halite has not been recognized in the studied samples?

Explanation: They are absent. This is also confirmed by the concentration of sedimentation brines.

Fig. 4. Please add some more detail and be more specific in the caption of these nice figures.

Explanation: We are not a very strong specialists in this issue to be sure about the definition. On A is classic pine pollen (Pinus sp., perhaps it is Pinus minuta Zaklinskaja). On B and D, obviously Carya, Taxodium (determined by tables of pollen of coniferous and broad-leaved trees presented on the Internet).

Line 203. “some periods”, please change with “some samples”.

Explanation: We are talking about the evolution of brines in the basin, therefore "in some periods".

Line 249-250. Not clear how glauberite is disposed in the samples. In this sentence I would rather state that glauberite is found either as isolated crystals within the bottom-grown halite or in distinct layers…. Please cite here Fig. 6.

Explanation: in bottom-grown halite, “chevron halite” – see Fig. 10.

Lines 254-257. I would rather reword like: “According to XRD analyses, anhydrite was detected within glauberite monocrystlas ….. “ Is it the correct meaning of this sentence?

Explanation: Calcium sulfate was detected both in glauberite single crystals (samples 6 and 9) and in water-insoluble halite residue (samples 2, 4, 14). In the water-insoluble residue of samples 2, 4, 14 - 100% anhydrite. In sample 1, 5, 6A, 7, 9A, 11, 13, only crystals of calcium sulfate are observed under the microscope from allogenetic minerals, but the material from these samples is not sufficient for XRD analysis.

Line 273 and following. In the method section, it was stated that the vapour (gas) phase was induced artificially by a low cooling. Here it is not clear if the described and anlyzed FI are two-phases already before the artificial cooling. This is rather important and could explain some “strange” homogenization temperature; if this is case (FI L+V in origin), I don’t think that the microthemometric data are realiable. Please explain.

Explanation: Regarding the features of gas phase formation in halite fluid inclusions, see [33, 64, 66]. Regarding the legitimacy of the use of such inclusions, see in Galamay A. R., Meng F., Bukowski K., Lyubchak A., Zhang Y., Ni P. 2019. Calculation of salt basin depth using fluid inclusions in halite from the Ordovician Ordos Basin in China. Geological Quarterly, , 63, 3, 619–628. http://dx.doi.org/10.7306/gq.1490

 Here is an excerpt from this post:

The question of the correspondence of the physicochemical data obtained from the study of fluid inclusions in halite to the conditions of the salt formation still remains controversial. The reason for this is the high plasticity and solubility of halite, for which a change in temperature or pressure in the salt column can lead to the development of such processes as stretching and the migration or destruction of previously arisen fluid inclusions (Holdoway, 1974; Roedder, 1984). However, the limits of permissible changes in P-T conditions at which fluid inclusions in halite retain their information content are still poorly studied.

Many physical and chemical observations (Petrichenko, 1973, 1988; Kovalevych, 1997; Bukowski et al., 2000; Galamay et al., 2003; Galamay and Bukowski, 2011; Meng et al., 2014) suggest that, after exposure of halite containing initially liquid inclusions, at temperatures ranging from 50°C to 110°C and with increased pressure, the informative value of primary inclusions in halite may be preserved. The gas phase can appear due to stretching or partial cracking in inclusions after heating to >50°C (Roedder, 1984), and the homogenization temperature of such inclusions corresponds to the temperature of salt deposit overheating (Petrichenko, 1973). Тhe integrity of early diagenetic inclusions with the primary fluid inclusions in the Majiagou Fm. halite have been preserved.  It indicates that they are of the same chemical composition, and that the secondary fluid inclusions have a different composition (Meng et al., 2018)”.

Line 296. This is confusing. I suggest to change”different zones of sedimentation textures” with in different halite lithofacies and in different portion of an individual lithofacies”

Explanation: The phrase "Thus, in our study, groups of inclusions from one zone of chevron halite were examined for each sample. Thus, in our study, groups of inclusions from one zone of chevron halite were examined for each sample." is correct. Regarding the importance of such an approach to the study of inclusions in halite, see [3]. This is often neglected, leading to misunderstandings in the interpretation of results.

Lines 324-325. It is not clear myself the relations between K decrease and the brine interaction with organics and/or clay mineral. Please explain.

Explanation: Experimental work on the change in the composition of seawater as a result of interaction with clay of continental origin (clay from the Moscow region) was carried out by A. N. Buneyev (1956). He proved that after two treatments of seawater with clay, along with a slight decrease in Na+ in the water, there was a complete disappearance of K+ ions from the solution, which were completely absorbed as an exchangeable base. This experiment can to some extent be seen as a model of the phenomena that occur in lakes and seas when interacting with substances brought in by wind or continental runoff (Lebedev, 1959). Thus, in nature, the phenomenon of K+ decrease in the Crimean lakes of Kzyl-Yar and Perekop (compared to its content in the Black Sea) is observed due to the absorption of K+ by silt and continental clays (Valyashko, 1962). Earlier, we found that the reason for the simultaneous decrease in potassium and partially sodium content in brines of the small Cretaceous Sakon Nakhon basin of Laos was due to continental factors [50]. Therefore, given the ratio of ions in the brines of the basin we studied, we believe that continental factors (the brine interaction with organics and/or clay mineral) were crucial for the decrease in potassium content. The sodium content in brines is not in all cases directly correlated with the potassium content, which is due to brine concentration and other factors.

Lines 455-464. Not clear this paragraph; I’m not sure about the meaning of the appearance of the gas bubble. Is it related to process of leaching, collapse of the inclusions walls..? Please explain. It seems that many inclusions have varied their initial volume (e.g presence of large vapour bubble) affecting the crystallization temperature.

Explanation: The size of the gas phase in fluid inclusions is directly correlated with the size of the inclusions. This applies both to the artificially formed gas phase as a result of cooling or freezing of halite, and to the gas phase in inclusions in halite that crystallized at a temperature higher than 35-42℃. Regarding the specifics of the study of the homogenization temperature of gas-liquid inclusions, we cite: [1, 4, 32-33, 50, 56, 65-67, 71].

Line 494. Which type of isotopic studies?

Explanation: Isotopic composition B, Br, Li, Cl of halites, C, O and Mg isotopic composition of carbonate minerals, Sr, O and S isotope analysis were made gypsum and anhydrite.

Lines 470-473. I do not agree. The obtained data are not indicative of shallow water, these mast be supporte da facies analyses.

Explanation: Facies determinations were investigated of the Tuz Gölü samples were published Karakaya et al. (2021) [5].

Lines 473-473. How the Authors have recognized the seasonality (spring-summer)? Is there any link with the pollen remains whitin fluid inclusion?

Explanation: Massive pollen finds in fluid inclusions are recorded in samples with a lower temperature of halite formation. These two factors, in our opinion, correspond to a separate period in the life (seasons) of the sedimentation basin.

Lines 547 and following. This point is not clear. Why the basin was very shallow? For instance in the the Dead Sea the temperature of halite nucleation ranges from 24° to 36° (see Sirota et al 2016 and following papers).

Explanation: Here we do not specify the specific depth in our research. The term "shallow water" for a salt-bearing basin is not precisely defined. In the literature, it means - from the first meters to the first tens of meters. Which layer (in meters) of brine reacts to changes in air temperature depends on many factors and requires special research. However, we have no confirmation by facies analysis, so we delete „insignificant” depth.

Comments corrected and incorporated into the text:

Line 18: bottom water layers, change with “ bottom waters”

Lines 24-25: I suggest rewording of this sentence: “The sulfate minerals ….. are found within halite crystals.”

Line 27. Not clear the reason to report such 3 ranges of temperature in the abstract. It should be clarified or changed with a single range.

Line 67. “Objectives”, three main objectives are reported.

Line 67. Delete “based on the obtained data”.

Line 70. “Depend” change with “depending on”

Line 99. I suggest to change “Then, due to” with “As a consequence of..”

Line 101. During the MSC… instead of “Concerning”…

Line 102. “They” is it referred to Akgun et ?

Lines 117-119. I suggest rewording: The Pliocene-quaternary succession consists of…. clay-rich deposits ()….

Fig. 1. Konya basins. This name is not specified in the main text.

Lines 124-127. These information are more adequate for the Material and Method section including coordinates of the studied wells.

Fig.2. In the legend of the map, the Paleocene-Eocene rocks are indicated as sedimentary rocks, other older rock are described with the name of the Formation. Please be consistent and make clear which type of sed. Rocks. Moreover, I suggest to change “Discordant” with “Unconformity”. In the well logs, a very general Miocene-Pliocene age is indicated; is there any more accurate time constrain allowing, for instance, for the identification of the Mio-Pliocene boundary?

Line 133. Ultramicrochemical method. I think that this method should be briefly described to the readers of the Journal and not only referred to a specific reference.

Line 139. Is it the concentration of the ions that decrease after 2-3 analyses? Please clarify

Line 141. Bunsen method: please add a reference.

Line 148. Immersion method. Again, this method should be better explained in this section; the reference provided here (and for many other citations in the manuscript) is in Russian and not readable for most of the readers.

Line 161. Which are the main characteristics of this chamber?

Line 169. “The size of the cumulate halite crystals reach…”

Line 177. I suggest to change “zones” with “bands” and “rhythm” with “cyclic pattern”.

Line 179. Change “zone” with “band”

Line 211. Change “and amount to” with “with amount of”

Line 214. Not clear which is mentioned study here.

Table. Please clarify the two numbers in the brackets (ie. sum of single ions). Sycilly: change with Sicily

Line 221-223. Not clear how the percentages of the individual ions were calculated. Please clarify

Table 2. “Janecke unit” not clear why it is indicated in the topmost line. Not present in Tab.1

Line 228. Columnar and isometric: not clear this habit of the crystals. Are they prismatic and tabular or elongated?

Line 234. Between zones of chevron: are these dissolution pipes?

Line 257. Installed: ???

Line 291. Why “sedimentary”? I suggest deleting

Line 296. This is confusing. I suggest to change”different zones of sedimentation textures” with in different halite lithofacies and in different portion of an individual lithofacies”

Line 297. See my comment in the table 3: be more specific on the site of analysis of the fluid inclusions (single band, a group of bands,.. )

Line 299. … as oberved in Messinian marine brines elsewhere (add citation)

Line 321-322. Data: change with “concentrations in the studied salt successions, the composition of the basin brine…”

Line 341. Change “triangular” with “ternary”

Line 351. Dissolved onshore, process commonly described as continental leaching of older evaporites

Line 367-369. Which type of glauberite is forming according to this process? I suggest to include in the discussion (here or elsewhere) a comparison with the lithofacies described by Salvany et al 2007 in the Ebro Basin.

Line 385. Bassanite

Lines 387-389. There are continuous citations to interesting processes, methods and petrofacies described in articles written in Russian or Ukrainian. I think that some more detail of these process should be reported in the text given that Minerals is an international journal and the official language is English.

Fig. 9. In the caption: I suggest to change sedimentary halite with bottom-grown halite, here and throughout the text.

Lines 391-392. Day-night period: this is an interpretation not a description of the figure; better: alternation of clear and dark bands.

Lines 395-399 and following. Consider to compare the glauberite lithofacies with those of Salvany et al 2007 (Sedimentology)

Line 463. Delete ”preparation of”

Reviewer 2 Report

The paper by Anatoliy R. Galamay et al. is geochemistry of brine and paleoclimate reconstruction during sedimentation of messinian salt in the Tuz Gölü Basin (Türkiye), based on fluid inclusions. Using an ultramicrochemical method, it was determined that the halogenesis in the Tuz Gölü Basin in Messinian corresponds to the sulfate type and the sulfate- magnesium subtype It was established by XRD and immersion methods that within fluid inclusions in halite, allogeneic calcium sulfate crystals are represented by gypsum, bassanite, and anhydrite. The study of halite homogenization temperatures indicates the small depth of the basin during sedimentation, and the study of bromine in halite indicates the variation of the basin's paleotectonic conditions and the partial or temporary separation of different sections of the broad basin from each other. I think the paper is worth publishing in a scientific review such as Minerals. In my opinion the paper can be accepted only after minor revision. Please see my suggestions below.

1. The abstract needs to be rewritten, and the first two paragraphs need to be put into the introduction. It mainly reflects the research method, research purpose and research significance of the article.

2. In addition, the English language is poor and some words and phrases are misused. Please examine the full manuscript.

3. In FIG. 6 evaporite minerals, mineral morphology characteristics should be described, and mineral types should be clearly marked on the picture

4. The measurement of fluid inclusion homogenization temperature needs to be marked with the instrument model and test unit. In addition, the heating rate of 1 degree per minute is too fast.

5. Most of the halite photos provided in this article are recrystaled halite, some of the original halite should be provided.

6. Add references to the latest research area.

7. The determination of boron content must be the inclusion composition data, the whole rock data does not have any significance.

8.The article is 502 lines, see Table 5, but there are only four tables in the whole article, please check it carefully.

Author Response

Thank you for your review.
We have made some corrections in the text, which we have highlighted in color. We hope that these corrections improve the quality of the article.

Reviewer 3 Report

This study on fluid inclusions in Tuz Gölü is quite interesting. Thank you for bringing this work to us. There are a few minor suggestions and corrections in the publication. Good luck.

Author Response

(The authors gave the same response as above.)
